# The Long-Term Effect of Adapting the Vertical Position of Implants on Peri-Implant Health: A 5-Year Intra-Subject Comparison in the Edentulous Mandible Including Oral Health-Related Quality of Life

**DOI:** 10.3390/jcm9103320

**Published:** 2020-10-16

**Authors:** Ron Doornewaard, Hugo de Bruyn, Carine Matthys, Ewald Bronkhorst, Stefan Vandeweghe, Stijn Vervaeke

**Affiliations:** 1Department Periodontology & Oral Implantology, Dental School, Faculty Medicine and Health Sciences, Ghent University, 9000 Ghent, Belgium; carine.matthys@ugent.be (C.M.); stijn.vervaeke@ugent.be (S.V.); 2Department of Dentistry, Radboud University Medical Center, Radboud Institute for Health Sciences, 6525 EX Nijmegen, The Netherlands; ewald.bronkhorst@radboudumc.nl; 3Department Reconstructive Dentistry, Dental School, Faculty Medicine and Health Sciences, Ghent University, 9000 Ghent, Belgium; stefan.vandeweghe@ugent.be

**Keywords:** bone loss, dental implant, overdenture, implant survival, peri-implantitis, soft tissue, split-mouth design, oral health-related quality of life, patient-reported outcome measures

## Abstract

Despite high success rates of dental implants, surface exposure may occur as a consequence of biologic width establishment associated with surgery. This prospective split-mouth study evaluated the effect of early implant surface exposure caused by initial bone remodeling on long-term peri-implant bone stability and peri-implant health. Additionally, Oral Health-Related Quality of Life (OHRQoL) was assessed by means of the Oral Health Impact Profile-14 (OHIP-14). Twenty-six patients received two non-splinted implants supporting an overdenture in the mandible by means of locators. One implant was installed equicrestally (control) and the second one was installed subcrestally, taking at least 3 mm soft tissue thickness into account (test). During initial bone remodeling (up to 6 months postoperatively), equicrestal placement yielded 0.68 mm additional surface exposure compared to subcrestal placement (*p* < 0.001). Afterwards, bone level and peri-implant health were comparable in both treatment conditions and stable up to 5 years. The implant overdenture improved OHRQoL (*p* < 0.01) and remained unchanged thereafter (*p* = 0.51). In conclusion, adapting the vertical position of the implant concerning the soft tissue thickness prevents early implant surface exposure caused by initial bone remodeling, but in a well-maintained population, this has no impact on long-term prognosis. The treatment of edentulousness with an implant mandibular overdenture improves OHRQoL.

## 1. Introduction

To provide functional comfort in the edentulous patient, an overdenture retained on two implants has been suggested as the first choice of treatment for the edentulous mandible [1]. The recent literature yields treatment success over 95% after 10 years of function [2]. Success could be determined by implant factors such as long-term peri-implant bone stability and the absence of inflammation in the peri-implant tissues or by patient factors such as the Oral Health-Related Quality of Life (OHRQoL). 

The effect of peri-implant mucosal tissue thickness on peri-implant bone stability has been described in animals and suggests a certain minimum width of peri-implant mucosa as a prerequisite, allowing a stable soft tissue attachment [3]. This was confirmed in humans and refined with the conclusion that a soft tissue thickness of 2 mm or less resulted in crestal bone loss up to 1.45 mm [4]. More recently, Vervaeke and co-workers concluded that the initial bone remodeling was affected by soft tissue thickness [5]. Furthermore, they suggested that an unforeseen exposure of the implant surface during initial bone remodeling should be avoided by adapting the vertical position of the implant during surgical placement in relation to the available preoperative soft tissue thickness. In the light of the hype that currently exists around peri-implantitis, it has been questioned whether the early exposure of implant surfaces to soft tissues could hamper peri-implant health or may pose a risk for the future development of peri-implantitis. Galindo-Moreno and co-workers concluded in an 18-month study that early implant surface exposure was predictive for additional bone loss [6]. Another clinical study, including 105 implants in 21 patients, concluded that initial bone loss and surface exposure at 2 years of function was identified as a predictor for further bone loss after 10 years of function [7].

Another subject of debate lies in the predictability of biologic peri-implant health parameters in relation to future risk for disease development or progression. Jepsen and co-workers could not demonstrate a difference in bleeding on probing between stable sites and sites with progressive bone loss [8]. However, bleeding on probing was characterized by a high negative predictive value, and thus an absence of inflammation can be an indicator for stable peri-implant conditions. In a long-term follow-up study of single implants functional for 16–22 years, Dierens and co-workers described very stable long-term bone stability with a 6% incidence of peri-implantitis. Despite this low incidence, 80% of the implants presented signs of inflammation with bleeding on probing [9]. Furthermore, they found a low correlation between probing pocket depth and bone levels. Hence, they concluded that probing depths are of limited value in predicting future peri-implant bone loss. Recently, based on 4951 implants, it was concluded that only profuse bleeding or suppuration did correlate with long-term bone loss, but no positive correlation was found for minimal bleeding and bone loss [10]. The above-mentioned findings of the clinical studies are in accordance with a recently published critical review by Doornewaard and co-workers [11]. This review included 41 articles representing 4198 patients initially treated with 9657 implants and showed the absence of a correlation between bone loss and the biologic parameters mean probing pocket depth and mean bleeding on probing. It needs to be mentioned that the outcomes of the latter study could have been biased by the fact that biologic parameters are reported very often in an inconsistent and incomplete manner. 

In addition to peri-implant health parameters, the success of an implant treatment should be determined by the Oral Health-Related Quality of Life (OHRQoL) [12]. In dentistry, the Oral Health Impact Profile-14 (OHIP-14) questionnaire is a widely used and validated instrument focusing on the impact of medical care on social and functional well-being [13]. 

Hence, the aim of this prospective split-mouth clinical study is to evaluate the long-term effect of adapting the vertical position of implants on peri-implant bone stability and peri-implant health, and secondarily to assess the oral health-related quality of life of patients restored with mandibular implant-retained overdentures. 

The short-term data regarding the peri-implant bone stability and peri-implant health were earlier published by Vervaeke and co-workers [5].

## 2. Experimental Section

### 2.1. Patient Population and Surgical/Prosthetic Procedures

This prospective split-mouth study included edentulous patients in need of a two-implant supported overdenture in the mandible. The patient selection, surgical, and prosthetic procedures have been described previously by Vervaeke and co-workers [5].

Patients received two dental implants (Astra Tech Osseospeed TX™, Dentsply implants, Mölndal, Sweden) inserted using a one-stage surgical procedure with an open flap. One control implant was installed equicrestally (group 1), according to the manufacturer’s guidelines. The vertical position of the test implant (group 2) was adapted to the soft tissue thickness, allowing at least 3 mm space for biologic width re-establishment. For example, if mucosal thickness was 2 mm, the test implast was installed 1 mm subcrestally. A systematic non-random assignment was applied to determine the position of the test and control implants by alteration of the experimental site for every consecutively included patient. If sufficient primary stability could be achieved, implants were immediately restored with locator abutments (Locator, ZEST Anchors LLC, Escondido). In the case of insufficient primary stability (<20 Ncm) in one or both implants, a two-stage protocol was preferred for both implants and were restored with locator abutments after 3 months. The crestal bone was slightly adapted around the subcrestally placed implant to install the locator abutments without direct contact between bone and abutment.

To achieve a balanced occlusion and articulation, appropriate teeth position, and appropriate smile line, all patients received new removable dentures in the mandible and maxilla before surgery. After surgery, the removable dentures were adapted to connect with the implants by one experienced prosthodontist (C.M.).

All patients were treated at the Ghent University Hospital by the same surgeon (S.V.) and prosthodontist (C.M.) between January 2013 and September 2014. Patient follow-up and supportive professional maintenance was done by two calibrated periodontists (S.V. and R.D.) and one prosthodontist (C.M.) for the technical follow-up. All patients were thoroughly informed and signed written informed consent, and the clinical trial has been conducted in full accordance with the Helsinki Declaration (1975) as revised in 2000. The ethical committee of the Ghent University Hospital approved the study protocol under registration number B670201215160.

### 2.2. Clinical and Radiographic Examination

The clinical and radiographic examination up to two years has been described previously by Vervaeke and co-workers [5]. Follow-up visits after surgery were planned at 1 week as well as at 1, 3, 6, 12, 24, 36, 48, and 60 months. Three months after surgery, when soft tissue healing was fully established, and during later control visits, peri-implant health was monitored by measuring probing pocket depths, bleeding on probing, and plaque scores on four implant sites: midmesial, middistal, midbuccal, and midlingual. Bleeding on probing and the presence of plaque were assessed on a dichotomous scale with 0 being absent and 1 being present. The scores were used to recalculate the parameters per implant. 

Digital peri-apical radiographs were taken immediately after implant placement (baseline) and after 3, 6, 12, 24, 36, 48, and 60 months using a guiding system in order to obtain the X-rays perpendicular to the film (Rinn XCP, Dentsply Sirona, Charlotte, NC, USA). The radiographs were calibrated using the length of the implant, the distance between the threads of the implant, or the diameter of the implant. Bone levels were determined as the distance from a reference point, which corresponds with the lower edge of the smooth implant bevel at the implant–abutment interface, to the most crestal bone-to-implant contact point. The bone loss is determined by the difference of the bone level directly after implant placement and the bone level at the follow-up visit.

If necessary, calculus and plaque were removed, and oral hygiene was reinforced during follow-up visits. Instructions with a (electric) toothbrush and interdental brushes were given based on the need, preferences, and dexterity or motoric skills of the patient.

The Oral Health Impact Profile-14 questionnaire (OHIP-14) was used to measure the change in oral health-related quality of life over time. It consists of 2 questions per domain scored using a Likert scale and capturing functional limitation, physical pain, psychological discomfort, physical disability, psychological disability, social disability, and handicap. Score 0 means no discomfort at all, and score 4 is indicative for a highly negative answer to the question. The total score of the questionnaire can range from 0 (maximally positive on all items) to 56 (maximally negative). The questionnaire was assessed before surgery as well as 3 and 60 months after connection of the prosthesis with the implant. The impact of the change was assessed by calculating the “effect size” with the use of the following formula: ((mean-OHIP before surgery)—(mean-OHIP three months after connection))/SD before surgery. As proposed by Cohen 1977, an “effect size” > 0.8 is interpreted as large, 0.6 is interpreted as moderate, and 0.2 is interpreted as small. 

### 2.3. Statistical Analysis

Data analysis was performed in SPSS Statistics 26 (SPSS Inc., Chicago, IL, USA). Outcomes are reported with descriptive statistics (mean, standard deviation (SD), median, range) and visualized through boxplot representation. All analyses concern pair-wise comparisons within patients. For dichotomous variables, the McNemar test was used, and for continuous variables, paired t-tests were applied. The 95% confidence intervals (95% CI) are given to show the precision of an estimate of a certain effect. The sample size was calculated using an SAS Power and Sample size calculator for related samples based on an effect size of 1 mm mean bone level difference between test and control and a standard deviation of 0.60, with the level of significance set at 0.05 and β = 0.80. The effect estimation was based on findings published previously [14].

An analysis of the measurement error for the continuous variable bone level between the observer S.V. and R.D. was performed by the use of a scatterplot representation and a paired-t-test. The random error, or duplicate measurement error (DME), was calculated with the formula s*d/√2*.

Incidence of peri-implantitis is based on the definition of peri-implantitis according to the 2017 Consensus report of the World Workshop on the classification of Periodontal and Peri-Implant Diseases and Conditions [15]. Implant success was defined in two ways: firstly, as 2 mm bone loss in combination with bleeding on probing as proposed by Klinge et al. [16], and secondly, as 1 mm additional bone loss after initial bone remodeling. 

## 3. Results

### 3.1. Study Population

Twenty-six patients were initially included in the study. One patient was excluded after starting smoking during the healing phase. In another patient with a knife-edge crest, both implants were installed subcrestally in order to have both implants completely surrounded by crestal bone. As a result of the absence of a control condition, this patient was excluded for further statistical analysis. Hence, 24 patients with two implants each (48 implants) were available for the 5-year follow-up. For 19 cases, the primary stability was high enough to use a one-stage protocol. In five patients, the primary stability required a two-stage submerged protocol. The baseline for these patients was the moment of abutment connection, which was approximately 3 months after implant placement.

The study population consisted of 13 men and 11 women with a mean age at implant placement of 65 years (SD = 9.38, range = 43–81). It was known that 16 out of the 24 patients had lost their teeth due to periodontal disease; for the other eight patients, the reason for tooth loss was unknown. Of the 24 included patients, only one patient could not attend the 3- and 4-year follow-up visit due to medical reasons, and another patient did not show up for the 4-year follow-up visit; however, all 24 patients attended the 5-year follow-up visit.

### 3.2. Survival Rate, Mean Bone Level Difference, and Mean Bone Loss

All implants were present after at least 5-years of follow-up, which resulted in a survival rate of 100%.

The analysis of the measurement error for bone level between the two observers (S.V. and R.D.) showed a mean difference of 0.024 with a 95% CI of between −0.0004 and 0.0484, resulting in a *p*-value of 0.054, which was indicative for no significant structural error. The standard error, or duplicate measurement error, was 0.046, which could be interpreted as low. The outcome of the structural error and random error are both indicative for a high inter examiner agreement.

The mean bone level and the corresponding changes for both placement protocols at baseline and after 6, 12, 24, 36, 48, and 60 months are shown in Table 1. A boxplot representation of the bone level for both treatment protocols at the subsequent time points is given in Figure 1. Initially, the bone level of the implants in both treatment protocols is comparable and basically located at the implant crest. At all other time points, a statistically significant bone level difference could be observed, all in favor of the subcrestally placed implants. 

In the first six months, bone remodeling was 0.7 mm for the equicrestally placed implants and 0.0 mm in the subcrestally placed implants. Six months is the time period considered appropriate for initial bone remodeling, following biologic width establishment. Figure 2 shows the mean bone loss between 6 and 60 months for both groups. For the equicrestally placed implants, this was −0.09 mm (SD 0.47) with a maximum additional bone loss of 0.92 mm. The negative number of the mean is indicative for a small but statistically and clinically irrelevant bone gain (*p* = 0.335). For the subcrestally placed implants, this change was 0.08 mm (SD 0.16) with a maximum loss of 0.48 mm after initial bone remodeling. Although this change was statistically significant (*p* = 0.021), it can be considered clinically irrelevant. When both treatment protocols are compared, the difference in bone loss between 6 and 60 months was not statistically significant (*p* = 0.077). Figure 3 is illustrative for the bone remodeling over time in both placement protocol, with the visible implant surface exposure in the equicrestally placed implant. 

### 3.3. Peri-Implant Health

After 60 months, the overall mean plaque score based on all implants was 0.39 (SD 0.35 range 0.00–1.00), with a mean plaque score of 0.39 for the equicrestally placed implants (SD 0.34, range 0.00–1.00) and 0.39 for the subcrestally placed implants (SD 0.37, range 0.00–1.00). At 60 months, the mean plaque score of equicrestally and subcrestally placed implants was not statistically significantly different (*p* = 1.00). The overall mean bleeding on probing off all implants was 0.18 (SD 0.24, range 0.00–1.00), with a mean bleeding on probing of 0.20 (SD 0.26, range 0.00–0.75) for the equicrestally placed implants and 0.16 (SD 0.23, range 0.00–1.00) for the subcrestally placed implants. At 60 months, the mean bleeding on probing of equicrestally and subcrestally placed implants was not statistically significantly different (*p* = 0.590). The overall mean probing pocket depth based on all implants was 2.04 mm (SD 0.53, range 1.00–3.25), with a mean probing pocket depth of 1.98 mm (SD 0.52, range 1.00–3.00) for the equicrestally placed implants and 2.09 mm (SD 0.55, range 1.25–3.25) for the subcrestally placed implants. At 60 months, the mean probing pocket depth between of equicrestally and subcrestally placed implants was not statistically significantly different (*p* = 0.257). 

### 3.4. Prevalence of Peri-Implantitis

According to the 2017 Consensus report of the World Workshop on the classification of Periodontal and Peri-Implant Diseases and Conditions, the incidence for peri-implantitis in both study populations is 0%. None of the implants showed bone levels ≥ 3 mm apical of the most coronal portion of the intraosseous part of the implant and/or probing pockets depths ≥ 6 mm. 

If a cross-sectional analyses after 5 years is performed and taking bone loss of 2 mm with bleeding on probing and/or suppuration to define disease as proposed by Klinge and colleagues [16], only one implant in the present study showed a bone level of more than 2 mm in combination with bleeding on probing (Table 2), resulting in a success of 97.9% of all implants, respectively 95.8% for the equicrestally and 100% for the subcrestally placed implants. 

When a longitudinal analysis is performed with bone loss over time, the maximum bone loss after initial bone remodeling was 0.92 mm for the equicrestal and 0.48 mm for the subcrestal treatment protocol. When considering 1 mm of bone loss after initial bone remodeling as a success, 100% of the implants in both treatment protocols were considered a success.

### 3.5. Oral Health-Related Quality of Life

The mean OHIP-14 score before surgery was 10.08 (SD 9.42, range 0–34). Three months after connection, the mean score reduced to 3.46 (SD 4.60, range 0–17); this reduction was statistically significant (*p* < 0.01) and indicative for an improvement in OHRQoL. The reduction was statistically significant for all seven domains, with a large effect size for physical pain and social disability. For the other domains, the effect size was moderate (Table 3). 

At 60 months, the mean OHIP-14 score was 4.33 (SD 5.92, range 0–15). Between 3 and 60 months, no statistically significant difference was observed (*p* = 0.51), which is indicative of a stable OHRQoL over time. 

## 4. Discussion

This prospective split-mouth clinical study evaluated the effect of long-term implant surface exposure, which is induced by biologic width re-establishment, on peri-implant bone stability and peri-implant health in patients treated with an implant-supported overdenture in the mandible. The applied split-mouth design corrects for inter-individual variability from the estimates of the treatment effect [17].

The difference in this study population in mean bone level between equicrestally and subcrestally placed implants at 6 months is 0.68 mm. The 95% confidence interval of the mean shows a 95% chance that the mean difference in the true population will be between 0.36 and 1.00 mm. Even the lower number of the mean difference of the true mean is already suggestive for clinically relevant differences in mean bone level. For all other time intervals, the same conclusion could be made. 

Compared to the short-term follow-up earlier published by Vervaeke and co-workers [5], no significant changes could be observed regarding peri-implant bone stability and peri-implant health when the 2-year data are compared with the 5-year data, which is indicative of stable peri-implant health over time. 

A recent systematic review and meta-analysis of 16 studies [18] concluded in a quantitative analysis that subcrestal and equicrestal implant placement yield comparable peri-implant bone loss. However, in the presence of a thin tissue, a subcrestal placement of the implant is preferred, because it may reduce the risk for implant exposure in the future, thus avoiding peri-implant pathologies. More studies suggested a certain minimum width of peri-implant mucosa as a prerequisite, allowing a stable soft tissue attachment [4,19,20,21,22]. The results of the present study are in agreement with the aforementioned papers. Hence, one should anticipate for the preferred biologic width establishment to prevent early implant surface exposure caused by initial bone remodeling by adopting implant depth positioning in relation to soft tissue thickness.

A recent clinical trial tried to overcome the initial bone remodeling due to biologic width re-establishment by using a soft tissue tenting technique [23]. These implants were placed equicrestal with soft tissue tenting over 2 mm healing abutments. The implants in the control group were placed 1.5 mm subcrestally. The bone loss between both groups was statistically significantly different and in favor of the subcrestally placed implants. They concluded that soft tissue tenting could increase soft tissue thickness. However, the latter technique is leading to greater bone loss compared to the subcrestal placement of the implants. Based on the present paper in line with the available evidence, it is advised to adapt the surgical position of the implant in relation to the available pre-operative soft-tissue thickness. This contradicts the protocols often advised by implant manufacturers suggesting that implant design features alone may prevent bone loss. 

Radiographic analysis of the subcrestally and equicrestally placed implants showed a minimal bone loss over time after the initial bone remodeling, although it was not clinically relevant. The findings of this paper are in accordance with earlier published papers, showing comparable results for peri-implant bone stability in patients treated with a two-implant overdenture in the mandible [24,25,26].

The present study demonstrated only small and clinically irrelevant differences for the biological parameters between equicrestally and subcrestally placed implants at all time intervals. Despite direct exposure of the implant threads, this did not lead to further bone loss, since there was no statistically significant difference in bone level between 6 and 60 months. One should keep in mind that all patients in the present study were fully edentulous and were compliant with oral hygiene. Whether this outcome is also valid in non-compliant patients is questionable as suggested by scarce evidence. It is highly unlikely that scientifically sound, randomized control trials in humans could be initiated in non-compliant patients given the unethical approach this would require. However, some evidence in the literature is in contradiction with the present finding. In partially edentulous patients, an early exposure of the implant surface was indicative for future bone loss [6]. It is tempting to suggest that partially edentulous patients harbor potentially more pathogenic peri-implant microflora explanatory for more bone loss in case of exposed implant surfaces [27]. Another 10-year follow-up study included 25 patients with an edentulous mandible restored with five implants and a fixed prosthetic rehabilitation. Not all of their patients complied with professional peri-implant maintenance therapy between year 3 and 10. Additionally, with a fixed prosthetic rehabilitation, maintaining a good oral hygiene was more demanding [7]. The positive effect of a regular peri-implant maintenance therapy has been described in a systematic review with meta-analysis by Monje and colleagues [28]. It is well understood that regular peri-implant maintenance therapy is mandatory to prevent biologic complications and ameliorates the long-term success rate.

As far as peri-implant health is concerned, the current findings are in accordance with other papers, which found no difference in BoP and/or PPD between equicrestally and subcrestally placed implants [29,30]. When the parameters of the mean bone level, bleeding on probing, and probing pocket depth are combined, only one implant in the present study showed a bone level of more than 2 mm in combination with bleeding on probing. However, a low probing pocket depth was scored for this implant, and the bone level stayed stable over time. The cross-sectional analysis to detect disease compared to the longitudinal analysis gave an overestimation for detecting disease. Despite a bone level above 2 mm after 60 months in combination with bleeding on probing, the bone loss after initial bone remodeling for this implant was below 1 mm, and the implant could be considered a success. 

It is questionable if the parameter mean, which is derived from four values per implant, is the best parameter to use for a statistical comparison of biologic parameters. This was also raised in the 5th EAO consensus conference where it was addressed that mean peri-implant bleeding scores and mean probing pocket depths are not adequate outcomes to measure health and disease. Frequency distributions of sites with a certain threshold of deep probing depths or sites demonstrating inflammation reflected by bleeding on probing are considered more appropriate [31]. The frequency distribution (Table 2) of the data from the current paper shows probing pocket depths, which are all indicative of peri-implant health. The findings of the weak correlation between biologic parameters and bone level are in accordance with the paper by Doornewaard and co-workers and indeed suggest that the single use of a periodontal index not combined with (ongoing) bone loss seems not to be a reliable indicator to measure the peri-implant health [11]. 

The outcome of the OHRQoL is in accordance with earlier published papers. All papers indicate the superiority of an implant-supported overdenture compared to a conventional complete denture regarding the quality of life [32,33,34,35]. Moreover, a recent published paper investigating the difference in OHRQoL between patients with an implant fixed complete denture and patients with an implant overdenture could not find a significant difference in OHIP score between the two treatment groups [36]. The above-mentioned findings confirm the McGill consensus statement where it is stated that an implant-retained overdenture is the first choice of treatment for the edentulous mandible. It could be concluded that if patients are well maintained, this treatment protocol yields high success rates regarding patient quality of life and peri-implant health. 

## 5. Conclusions

Within the limitations of this study, it can be concluded that adapting the vertical position of the implant in relation to the soft tissue thickness prevents early implant surface exposure caused by initial bone remodeling. In a well-maintained population with regular peri-implant maintenance therapy, the effect of early implant surface exposure caused by initial bone remodeling on peri-implant bone stability and biologic parameters seems to be limited after a follow-up of 5 years. 

## Figures and Tables

**Figure 1 jcm-09-03320-f001:**
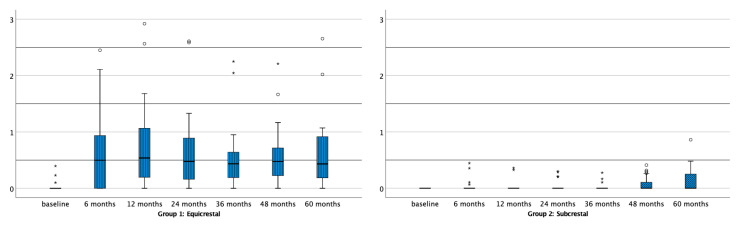
Boxplots representing bone level at subsequent time points for the equicrestally (Group 1) and subcrestally placed implants (Group 2). * Outliers (≥3× IQR above third quartile), ° suspected outliers (between 1.5 and 3× Inter Quartile Range above third quartile).

**Figure 2 jcm-09-03320-f002:**
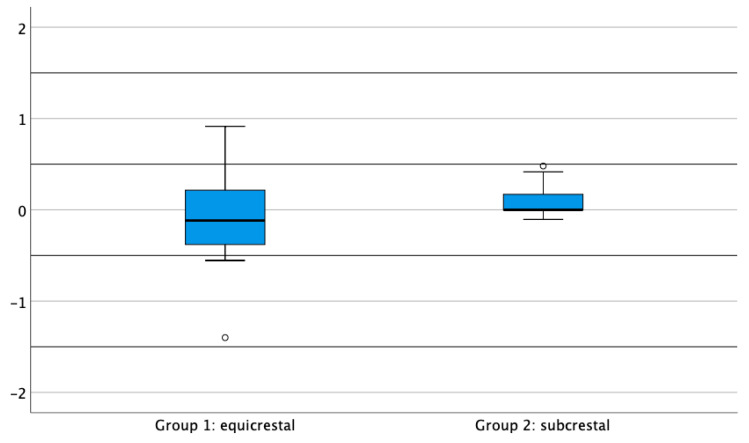
Boxplots representing bone level change between 6 and 60 months for the equicrestally (Group 1) and subcrestally placed implants (group 2). ° Suspected outliers (between 1.5× IQR and 3× IQR above third quartile), a negative number is indicative for bone gain.

**Figure 3 jcm-09-03320-f003:**
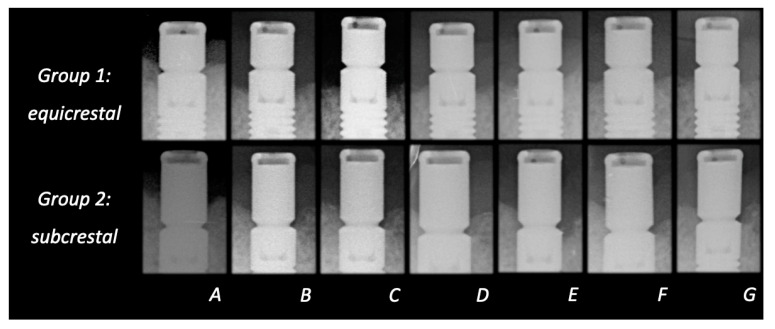
X-ray representing one and the same patients with the bone level directly after placement (**A**) and after 6 (**B**), 12 (**C**), 24 (**D**), 36 (**E**), 48 (**F**), and 60 (**G**) months for the equicrestally (Group 1) and subcrestally placed implants (Group 2).

**Table 1 jcm-09-03320-t001:** Mean bone level for each placement protocol and the bone level difference between respectively equicrestally and subcrestally placed implants; *p* is a result of a paired t-test comparing the bone level between placement protocols.

Bone Level									
Group 1: Equicrestal				Group 2: Subcrestal			Paired Difference		
	Mean (SD)	Median	Range	Mean (SD)	Median	Range	Mean Dif	95% CI	*p*
Baseline	0.03 (0.09)	0.00	(0.00–0.40)	0.00 (0.00)	0.00	(0.00–0.00)	0.030	(−0.009, 0.070)	0.123
6 months	0.72 (0.74)	0.59	(0.00–2.45)	0.04 (0.11)	0.00	(0.00–0.45)	0.678	(0.360, 0.996)	<0.001
12 months	0.78 (0.81)	0.54	(0.00–2.92)	0.03 (0.10)	0.00	(0.00–0.36)	0.746	(0.397, 1.096)	<0.001
24 months	0.69 (0.70)	0.51	(0.00–2.61)	0.04 (0.10)	0.00	(0.00–0.30)	0.644	(0.337, 0.951)	<0.001
36 months	0.59 (0.59)	0.44	(0.00–2.25)	0.04 (0.10)	0.00	(0.00–0.36)	0.549	(0.297, 0.802)	<0.001
48 months	0.56 (0.54)	0.46	(0.00–2.21)	0.07 (0.13)	0.00	(0.00–0.41)	0.487	(0.236, 0.737)	0.001
60 months	0.62 (0.66)	0.44	(0.00–2.66)	0.12 (0.22)	0.00	(0.00–0.86)	0.500	(0.219, 0.782)	0.001

SD, standard deviation; CI confidence intervals.

**Table 2 jcm-09-03320-t002:** Implant distribution at 5 years according to mean bone level and mean probing pocket depth; numbers between brackets show implants with bleeding on probing.

Probing Pocket Depth (mm)	Mean Bone Level (mm)						
	<0.5	0.5–0.99	1.00–1.49	1.50–1.99	2.00–2.49	≥2.5	total
≤1	1	1	0	0	0	0	2
1.1–2.0	17 (9)	2 (1)	3 (1)	0	0	1	23 (11)
2.1–3.0	17 (8)	4 (2)	0	0	1 (1)	0	22 (11)
3.1–4.0	1 (1)	0	0	0	0	0	1 (1)
4.1–5.0	0	0	0	0	0	0	0
>5.0	0	0	0	0	0	0	0
**total**	36 (18)	7 (3)	3 (1)	0	1 (1)	1	48 (23)

**Table 3 jcm-09-03320-t003:** Mean Oral Health Impact Profile (OHIP) score and the mean difference for each of the seven domains before surgery and three months after connection with the calculated effect size.

Domain	Mean OHIP (SD)		Paired Difference			Effect Size
	Before Surgery	3 Months After Connection	Mean Dif	95% CI	*p*	
functional limitation	2.04 (1.90)	0.79 (1.10)	1.25	(0.431, 2.069)	0.004	0.66
physical pain	3.13 (2.23)	0.88 (1.45)	2.25	(1.206, 3.294)	<0.001	1.01
psychological discomfort	2.50 (2.81)	0.54 (1.64)	1.96	(0.592, 3.325)	0.007	0.70
physical disability	1.63 (2.02)	0.29 (0.62)	1.34	(0.492, 2.175)	0.003	0.66
psychological disability	2.00 (2.17)	0.42 (0.78)	1.58	(0.506, 2.661)	0.006	0.73
social disability	1.45 (1.47)	0.13 (0.45)	1.32	(0.655, 2.012)	<0.001	0.90
handicap	1.33 (1.52)	0.33 (0.76)	1.00	(0.353, 1.647)	0.004	0.66

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
