# Peer review of "The Long-Term Effect of Adapting the Vertical Position of Implants on Peri-Implant Health: A 5-Year Intra-Subject Comparison in the Edentulous Mandible Including Oral Health-Related Quality of Life"

_jcm, 2020, doi:10.3390/jcm9103320_

Round 1

Reviewer 1 Report

The reserch is  a mere repetition of what is already known.

The design of this implant with platform swiching and micro threads is for subcrestal use, not yuxtacrestal. Bone resorption is expected always.

A regular peri-implant maintenance therapy is difficult with microthreads exposed.

Author Response

Dear Reviewer 1. First of all thank you very much for your time to review this manuscript. Below you find the answers to your comments.

The reserch is a mere repetition of what is already known.

We agree with the reviewer this is a continuation of an existing study. However, to the best of our knowledge, 5 year data on an intra-subject comparison evaluating the risk of early bone loss for peri-implantitis is not available in the literature.

The design of this implant with platform swiching and micro threads is for subcrestal use, not yuxtacrestal. Bone resorption is expected always.

Based on the available literature, standardized subcrestal implant placement is not recommended. Implant position should anticipate biologic width formation. In cases with adequate soft-tisue dimensions yuxtacrestal implant placement will not result in bone loss.

A regular peri-implant maintenance therapy is difficult with microthreads exposed.

Indeed we agree with reviewer 1 that peri-implant maintenace therpay is difficult if microthreads are exposed. In this paper we do have a prospective follow-up of patients with micothreads exposed and we can concluded that with a regular patient based maintenance therapy, the exposure of microthreads do not seem to have a negative impact on the long term outcome. So firstly we draw a conclusion that exposure could be prevented if the soft tissue thickness is taken into account. And secondly if the surface is exposed, and if there is a good regular maintenace therapy problems could be prevented. So hopefully this answer could also contribute to the first comment of the reviewer that in our perspective this long-term is of clinical relevance.

The PDF with changes is attached.

Kind regards

Reviewer 2 Report

1. Please explain more clearly in the methodology section, how much subcrestal the implants were placed in the test group. Was there a certain depth (in mm) below the alveolar crest where the implants were placed?

2. The test group showed a minor changes in the bone level. Discuss the limitations of measuring such small changes. 

3. Include the details of how the bone level changes were measured e.g. who did the measurements? were there any inter- or intra-observer variation/calibration?

Author Response

Dear Reviewer 2,

I want to thank you for your time to review this article. Your comments are a positive contribution to this article. Hopefully, the answers to your questions/comments meet your expectations. The PDF with track changes to the article is also attached.

Kind regards,

Ron 

  1. Please explain more clearly in the methodology section, how much subcrestal the implants were placed in the test group. Was there a certain depth (in mm) below the alveolar crest where the implants were placed?

The text is changed and a sentence is added to make this point more clear (Line 114-118):

The vertical position of the test implant (group 2) was adapted to the soft tissue thickness, allowing at least 3 mm space for biologic width re-establishment. . For example, if mucosal thickness was 2 mm the test implast was installed 1 mm subcrestally

  1. The test group showed a minor changes in the bone level. Discuss the limitations of measuring such small changes. 

Indeed the test group showed minor changes. The use of an x-ray to measure bone level/bone loss could give an underestimation of the true problem. Due to the repetition of the measurement between the two observers it was possible to obtain an indication of the structural and random error. Both are low, which are indicative of a high inter-examiner agreement. It also makes clear that even with minor changes the observers had an agreement and the x-ray seems to be reliable to measure this small changes.

On the other hand we need to admit that the gold standard to determine bone loss would be raising a flap and measure the bone loss/ bone level. However this is not ethical. Due to the lack of the gold standard we do not want to give an ICC but rather give the outcome of the structural error and random error. With these outcomes we could make an interpretation that even measuring small changes seems to be reliable enough to draw a conclusion.

  1. Include the details of how the bone level changes were measured e.g. who did the measurements? were there any inter- or intra-observer variation/calibration?

The next sentences are added:

            M&M line 203-205: An analysis of the measurement  error for the continuous variable bone level between the observer S.V. and R.D. was performed by the use of a scatterplot represantation and a paired-t-test. The random error, or duplicate measurement error (DME) was calculated with the formula: sd/√2​

            Results line 231-243: The analysis of the measurement error for bone level between the two observers (S.V. and R.D.) showed a mean difference of 0.024 with a 95% CI of between -0.0004 and 0.0484, resulting in a p-value of  0.054; indicative for no significant structural error. The standard error, or duplicate measurement error, was 0.046, which could be interpreted as low.. The outcome of the structural error and random error are both indicative for a high inter examiner agreement.

Reviewer 3 Report

This split-mouth prospective clinical study aimed to assess the effect of early implant surface exposure on long-term peri-implant bone stability and peri-implant health. In this study, 26 edentulous patients were included. For each patient, one implant was placed at bone level (equicrestal; control) and one implant was placed sub-crestal allowing least 3 mm of soft tissue thickness over the implant (test).  Peri-implant bone level, peri-implant health parameters, and Oral Health-Related Quality of Life (OHRQoL) were assessed up to five years post-operatively. They found significant differences between the two implant placement protocols in terms of peri-implant bone level at all follow-up visits. However, the difference in bone loss between 6 and 60 months was not statistically significant between the two groups. No other significant differences were found between the two groups for any other variables. It was concluded that adapting the vertical position of the implant concerning the soft tissue thickness prevents early implant surface exposure but in a well-maintained population this has no impact on long-term prognosis.

In general, it is a well-designed and well-written study and the paper can be of interest to readers of the Journal of Clinical Medicine. However, the study needs minor major revisions that are detailed below:

1) This paper appears to be a longer follow-up (5-year) on the following previously published study:

  • Vervaeke, S.; Matthys, C.; Nassar, R.; Christiaens, V.; Cosyn, J.; De Bruyn, H. Adapting the vertical position of implants with a conical connection in relation to soft tissue thickness prevents early implant surface exposure: A 2-year prospective intra-subject comparison. Journal of clinical periodontology 2018, 45, 605-612, doi:10.1111/jcpe.12871.

However, currently, it is not mentioned in the manuscript.  Authors must make it clear if it is a longer follow-up on a previously published study.

2) The study did not really fully address the study aim which is stated as” to evaluate the effect of long-term implant surface exposure induced by biologic width re-establishment on peri-implant bone stability and peri-implant health…”. To address this aim, all implants in the test group should have exposed surface at baseline and all implants in the control group should not have exposed surface. However, the main difference between the test and control group in the present study is their vertical placement level (equicrestal vs sub-creastal). Hence, the current aim of the study needs to be rewritten. I suggest that authors changed their aim to “… to evaluate the long-term effect of adapting the vertical position of implants on peri-implant health …”

3) The title needs to be changed as well according to the above mentioned comment

4) The term “implant surface exposure” can cause confusion. Implants with exposed surface usually refer to implants with both peri-implant hard and soft tissue loss. In the present study, authors refer to implant with peri-implant bone loss. I suggest that authors use the term “peri-implant bone loss” instead.

5) Page 3 line 96-97: “ A systematic random assignment was applied to determine the position of test and control implants by alteration of the experimental site for every consecutively included patient.” This is not a method of randomization. Please change “random” to “non-random”

6) Page 3 line 97-“If primary stability could be achieved” please change this sentence to “ If sufficient primary stability …”

7) Please add a paragraph on sample size calculation

Author Response

Dear reviewer 3

I want to thank you for your time to review this article. Your comments are a positive contribution to this article. Hopefully, the answers to your questions/comments meet your expectations. The PDF with track changes to the article is also attached.

Kind regards,

Ron 

1) This paper appears to be a longer follow-up (5-year) on the following previously published study:

  • Vervaeke, S.; Matthys, C.; Nassar, R.; Christiaens, V.; Cosyn, J.; De Bruyn, H. Adapting the vertical position of implants with a conical connection in relation to soft tissue thickness prevents early implant surface exposure: A 2-year prospective intra-subject comparison. Journal of clinical periodontology 2018, 45, 605-612, doi:10.1111/jcpe.12871.

However, currently, it is not mentioned in the manuscript.  Authors must make it clear if it is a longer follow-up on a previously published study.

We added the following sentence line 86-87:

The short-term data regarding the peri-implant bone stability and peri-implant health are earlier published by Vervaeke and co-workers [5].

2) The study did not really fully address the study aim which is stated as” to evaluate the effect of long-term implant surface exposure induced by biologic width re-establishment on peri-implant bone stability and peri-implant health…”. To address this aim, all implants in the test group should have exposed surface at baseline and all implants in the control group should not have exposed surface. However, the main difference between the test and control group in the present study is their vertical placement level (equicrestal vs sub-creastal). Hence, the current aim of the study needs to be rewritten. I suggest that authors changed their aim to “… to evaluate the long-term effect of adapting the vertical position of implants on peri-implant health …”

New aim line 82-85: Hence,the aim of this prospective split-mouth clinical study is to evaluate the long-term effect of adapting the vertical position of implants on peri-implant bone stability and peri-implant health, and secondary to assess the oral health-related quality of life of patients restored with mandibular implant-retained overdentures.

3) The title needs to be changed as well according to the above mentioned comment

New Title: The long-term effect of adapting the vertical position of implants on peri-implant health: a 5-year intra subject comparison in the edentulous mandible including Oral Health Related Quality of Life.

4) The term “implant surface exposure” can cause confusion. Implants with exposed surface usually refer to implants with both peri-implant hard and soft tissue loss. In the present study, authors refer to implant with peri-implant bone loss. I suggest that authors use the term “peri-implant bone loss” instead.

In the earlier published study by Vervaeke et al we also used the term implant surface exposure. For this reason we would like to use the same terminology. We would like to change the term to “ early implant surface caused by initial bone remodelling”

5) Page 3 line 96-97: “ A systematic random assignment was applied to determine the position of test and control implants by alteration of the experimental site for every consecutively included patient.” This is not a method of randomization. Please change “random” to “non-random”

Sentence changed to:  A systematic non-random assignment was applied to determine the position of test and control implants by alteration of the experimental site for every consecutively included patient

6) Page 3 line 97-“If primary stability could be achieved” please change this sentence to “ If sufficient primary stability …”

Sentence changed to: If sufficient primary stability could be achieved (insertion-torque ≥ 25 Ncm) implants were immediately restoredwith locator abutments (Locator,ZESTAnchorsLLC,Escondido).

7) Please add a paragraph on sample size calculation

The sample size was calculated using SAS Power and Sample size calculator for related samples based on an effect size of 0.6 mm and a standard deviation of 0.60, with the level of significance set at 0.05 and β = 0.80. The effect estimation was based on findings Vervaeke et al. 2014.

We decided to use 95% confidence intervals. They show the reader the precision of an estimate of a certain effect. And especially in case of non significant findings it show, to what extent that is due to lack of effect or a lack of precision. Hence, we provided 95% ci’s for all our estimates.

Reviewer 4 Report

The article describes an interesting topic with a well-designed prospective study. 

I just have some comments and questions for Authors regarding the Experimental section:

  • Line 97-99 This part need more details. It is not specified what happened with 1) an implant that resulted with an insertion torque between 20 and 25 Ncm. 2) If the two implants achieved different insertion torque. Please give more details. 
  • Line 137. There might be a typo, where "in" should be replaced by "is"
  • Line 148. Student t-test assumes the normality distribution of the variable (which is not obvious, especially for the subcrestal group). How was tested the normality distribution of each variable?
  • Line 149-153. The result of the sample size calculation is not reported. Please comment. 

Author Response

Dear reviewer 4

I want to thank you for your time to review this article. Your comments are a positive contribution to this article. Hopefully, the answers to your questions/comments meet your expectations. The PDF with track changes to the article is also attached.

Kind regards,

Ron

The article describes an interesting topic with a well-designed prospective study. 

I just have some comments and questions for Authors regarding the Experimental section:

  • Line 97-99 This part need more details. It is not specified what happened with 1) an implant that resulted with an insertion torque between 20 and 25 Ncm. 2) If the two implants achieved different insertion torque. Please give more details. 

We changed the sentences to make it more clear:

If sufficient primary stability could be achieved implants were immediately restoredwith locator abutments (Locator,ZESTAnchorsLLC,Escondido). In case of insufficient primary stability (< 20 Ncm) in one or both implants, a two-stage protocol was preferred for both implants and were restored with locator abutments after 3 months.

  • Line 137. There might be a typo, where "in" should be replaced by "is"

Score 0 means no discomfort at all and score 4 is indicative for a highly negative answer to the question.

  • Line 148. Student t-test assumes the normality distribution of the variable (which is not obvious, especially for the subcrestal group). How was tested the normality distribution of each variable?

For our outcomes (each of which is in itself the difference of “after and before” and as a result of that already much closer to a normal distribution then a cross-sectional observation), a sample size of more than 20 is enough to use t-tests. The normality was not formally tested with a test like the Kolomogorov-Smirnov test. Normality tests have low power and can be considered to be a “easy” way to get around distribution issues in small studies. Instead we looked at boxplots for a visual inspection of the distributions.

  • Line 149-153. The result of the sample size calculation is not reported. Please comment. 

The sample size was calculated using SAS Power and Sample size calculator for related samples based on an effect size of 0.6 mm and a standard deviation of 0.60, with the level of significance set at 0.05 and β = 0.80. The effect estimation was based on findings Vervaeke et al. 2014.We decided to use 95% confidence intervals. They show the reader the precision of an estimate of a certain effect. And especially in case of non significant findings it show, to what extent that is due to lack of effect or a lack of precision. Hence, we provided 95% ci’s for all our estimates.

Round 2

Reviewer 1 Report

No added comments